# SARS-CoV-2 Papain-like Protease Negatively Regulates the NLRP3 Inflammasome Pathway and Pyroptosis by Reducing the Oligomerization and Ubiquitination of ASC

**DOI:** 10.3390/microorganisms11112799

**Published:** 2023-11-17

**Authors:** Huan Meng, Jianglin Zhou, Mingyu Wang, Mei Zheng, Yaling Xing, Yajie Wang

**Affiliations:** 1Department of Clinical Laboratory, Beijing Ditan Hospital, Capital Medical University, Chaoyang District, Beijing 100015, China; 2Bioinformatics Center of Academy of Military Medicine Science, Beijing 100850, China

**Keywords:** *SARS-CoV-2*, papain-like protease, NLRP3 inflammasome, IL-1β, pyroptosis, ASC, deubiquitination

## Abstract

The interaction of viruses with hosts is complex, especially so with the antiviral immune systems of hosts, and the underlying mechanisms remain perplexing. Infection with *SARS-CoV-2* may result in cytokine syndrome in the later stages, reflecting the activation of the antiviral immune response. However, viruses also encode molecules to negatively regulate the antiviral immune systems of hosts to achieve immune evasion and benefit viral replication during the early stage of infection. It has been observed that the papain-like protease (PLP) encoded by coronavirus could negatively regulate the host’s IFNβ innate immunity. In this study, we first found that eight inflammasome-related genes were downregulated in CD14+ monocytes from COVID-19 patients. Subsequently, we observed that *SARS-CoV-2* PLP negatively regulated the NLRP3 inflammasome pathway, inhibited the secretion of IL-1β, and decreased the caspase-1-mediated pyroptosis of human monocytes. The mechanisms for this may arise because PLP coimmunoprecipitates with ASC, reduces ASC ubiquitination, and inhibits ASC oligomerization and the formation of ASC specks. These findings suggest that PLP may inhibit strong immune defenses and provide the maximum advantage for viral replication. This research may allow us to better understand the flex function of CoV-encoding proteases and provide a new perspective on the innate immune responses against *SARS-CoV-2* and other viruses.

## 1. Introduction

*Severe acute respiratory syndrome coronavirus 2* (*SARS-CoV-2*) is a single-strand RNA virus. The proteins encoded by *SARS-CoV-2* genomes can participate in the replication, transcription, and maturation of the virus, interact with host proteins, and modulate host antiviral immune responses [1].

Papain-like protease (PLP) is encoded by coronavirus’s non-structural protein 3 (nsp3), which exhibits catalytic activity in processing the cleavage of nsp1↓nsp2, nsp2↓nsp3, and nsp3↓nsp4; participates in virus replication in conjunction with other proteases, such as 3C-like protease (chymotrypsin-like protease, 3CL); and recognizes the C-terminus of ubiquitin acting as a deubiquitinase [2,3]. Moreover, PLP has been reported to have additional functions. Previous studies in our laboratory showed that the coronavirus (CoV) PLP acts as an interferon-β (IFNβ)-antagonistic component [4]. Membrane-associated PLP (PLP-TM) of *NL63* or *severe acute respiratory syndrome coronavirus* (*SARS-CoV*) inhibited the stimulator of the interferon gene (STING)-mediated innate immune pathway, antagonized the nuclear translocation activation and the promoter-dependent induction of IFN regulatory factor 3 (IRF-3), and also negatively regulated the assembly of the STING mitochondrial antiviral signaling protein (MAVS)-TANK binding kinase 1 (TBK1)/TANK-binding kinase 1 (IKK) complex required for IRF-3 activation, thereby inhibiting IFNβ expression [5,6]. Moreover, *Middle East respiratory syndrome-related coronavirus* (*MERS-CoV*) PLP also antagonizes IFNβ-related immune responses. CoV PLP interacts with Beclin 1 to induce incomplete autophagy, thus impeding antiviral innate immunity [7]. Recent research indicates that *SARS-CoV-2* PLP shows deISGylating activity and catalytic activity on lysine(K)48-linked Ub (Ubiquitin) chains [8,9,10]. Moreover, as with *SARS* PLP and *MERS* PLP, *SARS-CoV-2* PLP also attenuates the IFNβ-related immune responses, and regulates the replication and spread of the virus [11].

IFN and other inflammatory cytokines such as IL-1β play essential roles in engaging innate immune defenses. The NLRP3 inflammasome is a common platform for the maturation of IL-1β and induces cell pyroptosis [12]. The NLRP3 inflammasome is composed of three key molecules including NLRP3, the adaptor apoptosis-associated speck-like protein containing a caspase recruitment domain (ASC), and pro-caspase-1. With the right stimuli, these three molecules can form a complex that further acts on downstream pro-inflammatory cytokines, such as IL-1β, and cleaves gasdermin D (GSDMD, a member of the gasdermin family) into GSDMD-N(the N-terminal of GSDMD) and GSDMD-C (the C-terminal of GSDMD), which subsequently mediate cell pyroptosis [12]. The oligomerization of ASC results in the formation of ASC specks, which act as a signal amplification mechanism for inflammasomes [13], while the ubiquitination of ASC is required for NLRP3 inflammasome activation [14].

In most patients with severe COVID-19, the serum cytokines including IL-6, IL-2, IL-10, and TNF-α were elevated; however, there was no significant change in the secretion of IL-1β in asymptomatic patients and those with mild infections in the early stage, in some cases [15,16]. These phenomena indicated the complexity of *SARS-CoV-2*’s modulation of cytokine networks. Another study showed that *MERS-CoV* infection in cells also showed the non-efficient production of cytokines such as IL-1 and IL-18 [17]. We have found that coronavirus PLP can inhibit the production of IFNβ and induce autophagy in host cells. Moreover, it has been reported that IFNβ and IL-1β can regulate each other [18,19] and there is a cross-regulation between autophagy and the inflammasome pathway [20]. Based on this knowledge, this research focuses on the patterns whereby *SARS-CoV-2* PLP regulates the NLRP3 inflammasome pathway. The results showed that *SARS-CoV-2* PLP negatively regulated the NLRP3 inflammasome pathway, inhibited the secretion of IL-1β, and decreased the caspase-1-mediated pyroptosis of human monocytes. The related mechanisms may be related to the fact that PLP interacts with ASC, and, as a result, PLP interrupts the ASC–caspase-1 complex. What’s more, PLP reduces the oligomerization of ASC and the formation of ASC specks and reduces the ubiquitination of ASC. These findings suggest that CoV may encode PLP to regulate the cytokine network via complex mechanisms, affect the overall balance of host immunity, and achieve long-term coexistence with the host by acting as an immune evasion mechanism.

## 2. Materials and Methods

### 2.1. Reagents

The following reagents were acquired: Dulbecco’s Modified Eagle Medium (Gibco, Waltham, MA, USA, C11995500BT), rich culture medium RPMI 1640 (Sigma, St. Louis, MO, USA, R8758-500ML), FBS (Beyotime, Nantong, China, C0234), lipopolysaccharides from Escherichia coli O55:B5 (Sigma, 12190801), nigericin (Invivogen, San Diego, CA, USA, tlrl-nig), Phorbol 12-myristate 13-acetate (PMA) (Sigma, P1585-10mg), disuccinimidyl suberate (DSS) (Thermo Fisher Scientific, Waltham, MA, USA, A39267), an IL-1β ELISA kit (Dakewe, Shenzhen, China, 1110122), anti-DDDDK-tag mouse mAb (MBL, Tokyo, Japan, M185-3L), anti-Myc-tag mouse mAb (MBL, M192-3), anti-HA-tag mouse mAb (MBL, M180-3), anti-Myc-tag rabbit pAb (MBL, 562), NLRP3 (D4D8T) rabbit mAb (CST, Beverly, MA, USA, 15101S), V5-tag (D3H8Q) rabbit mAb (CST, 13202S), cleaved-IL-1β (Asp116) (D3A3Z) rabbit mAb (CST, 83186S), IL-1β (D3U3E) rabbit mAb (CST, 12703S), ASC/TMS1 (E1E3I) rabbit mAb (CST, 13833S), caspase-1 (D7F10) rabbit mAb (CST, 3866), gasdermin D (E8G3F) rabbit mAb (CST, 97558S), cleaved caspase-1 (Asp297) (D57A2) rabbit mAb (CST, 4199S), anti-cleaved N-terminal GSDMD antibody [EPR20829-408] (Abcam, Cambridge, UK, ab215203), β-Actin rabbit mAb (high dilution) (ABclonal, Woburn, MA, USA, AC026), anti-IgG (H + L chain) (mouse) pAb-HRP (MBL, 330), anti-IgG (H + L chain) (rabbit) pAb-HRP (MBL, 458), protein A + G agarose (Beyotime, P2012), lipofectamine 2000 reagent (Invitrogen, Waltham, MA, USA, 11668-019), CytoTox 96 Non-Radioactive Cytotoxicity Assay kit (Promega, Madison, WI, USA, G1780), a protease inhibitor cocktail (Beyotime, P1005), and NCM Western BlotStripping Buffer(New Cell & Molecular Biotech Co., Ltd, Suzhou, China, WB6200).

### 2.2. Reanalysis of Single-Cell RNA Sequencing Data

The method used for analysis is the same as in the previous study [21]. In brief, the raw scRNA-seq FASTQ files of the PBMCs were downloaded from the Genome Sequence Archive of the Beijing Institute of Genomics (BIG) Data Center (HRA000150 [dataset]. 23 April 2020. Full Monthly Release, 25 June 2020 [cited 11 October 2022]. Available from: https://ngdc.cncb.ac.cn/gsa-human/browse/HRA000150). These reads were then parsed using the cellranger (v.4.0.0) count pipeline with the human reference genome (GRCh38) to produce gene expression matrices. The subsequent analyses were carried out using R (v.4.0.2) scripts with the Seurat (v.3.2.2) package, as described in the NI paper [22]. Briefly, the gene expression matrices from the cellranger output were further individually filtered according to the parameters in the NI paper [22]. Then, using the “standard workflow” as specified at https://satijalab.org/seurat/v3.2/integration.html (accessed on 4 April 2021), an integrated and unbatched dataset was generated from different samples under four conditions. The integrated dataset was scaled before principal component analysis (PCA) was performed. The first 20 principal components (PCs) were used to build an SNN network and the Louvain algorithm was applied to the cluster cells with a parameter resolution = 1.4. Based on the chosen classic marker [22], the clusters were then categorized and annotated. Specifically, CD14+ monocytes (CD14 + mono; LYZ + CD14+), CD16+ monocytes (CD16 + mono; LYZ + FCGR3A+), monocyte-derived dendritic cells (mono DCs; CD1C+), plasmacytoid dendritic cells (pDCs; LILRA4+), and natural killer (NK) cells (KLRF1+) were selected and extracted to perform the subsequent analyses. Lastly, the cell clustering results were visualized in 2D space using the UMAP algorithm (Appendix A). Eight genes (*NLRP3*, *NLRP6*, *PYCARD*, *IL1B*, *IL18*, *TRIM31*, *FBXL2*, and *MARCH7*) were used to define the inflammasome pathway score.

### 2.3. Cell Culture and Stimulation

The human embryonic kidney HEK-293T, human bronchial epithelioid BEAS-2B cells, and human monocyte cells THP-1 were obtained from the Cell Resource Center, Peking Union Medical College (PCRC). The HEK-293T cells and BEAS-2B cells were cultured in Dulbecco’s Modified Eagle’s Medium (DMEM) supplemented with 10% FBS and 1% penicillin and streptomycin (P/S). The THP-1 cells were cultured in RPMI-1640 supplemented with 10% FBS and 1% P/S at 37 °C under 5% CO_2_. The THP-1 cells were differentiated into macrophages via treatment with 20 ng/mL PMA for 36 h. The THP-1 macrophages were then stimulated using 1 μg/mL LPS for 3 h followed by 10 μM nigericin for 1.5 h. The BEAS-2B cells were stimulated using 10 μg/mL LPS for 12 h followed by 10 μM nigericin for 3 h. The cell lysates were collected for Western blotting, and the supernatants were collected for ELISA.

### 2.4. Plasmid Construction and Transfection

The DNA sequences encoding the PLP-TM (amino acids (aa) 1564-aa2394 in pp1ab) of *SARS-CoV-2* Wuhan-Hu-1 (GenBank accession number NC_045512.2) were synthesized using GenScript (Nanjing, China) and were encoded into the V5/HisB vector between BamHI and EcoRI. The Myc-NLRP3, Myc-ASC, and Myc-pro-caspase-1 were provided by Jian Wang (Beijing Proteome Research Center, Beijing, China). The ASC was cloned into a pcDN3.1 HA-tagged vector or Flag-tagged vector (obtained from miaolingbio) between Hind III and EcoR I. The cDNAs encoding the human pro-IL-1β were obtained via the reverse transcription of total RNA from the THP-1 macrophages, followed by PCR using specific primers (Forward primer: GAAT G GATCC GCCACC ATGGCAGAAGTACCTGAG, reverse primer: GCCG G AATTC GGAAGACACAAATTGCATGGTGAAG), and the pro-IL-1β was encoded into the pcDN3.1 Flag-tagged vector between BamHI and EcoRI. The Flag-tagged vector contains the CMV promoter. According to the operation manual, Lipofectamine 2000 was used to transfect the plasmids into cells.

### 2.5. Lentivirus Production and Infection

The GFP-Flag-PLP-TM-Lentivirus and GFP-CT-Lentivirus (control vector) were constructed by Genechem (Shanghai, China). Then, 5 × 10^8^ TU/mL GFP-Flag-PLP-TM-Lentivirus or 1 × 10^8^ TU/mL GFP-CT-Lentivirus was used to infect the THP-1 cells for 48 h, after which the infected THP-1 cells were cultured using 2.5 μg/mL puromycin for 72 h.

### 2.6. Co-IP Assay and Co-IP Ubiquitination Assay

The plasmids were transfected into HEK-293T cells, as indicated, for 24 h. The cells were lysed using a NP-40 lysis buffer (50 mmol/L Tris-HCl pH 7.4, 150 mmol/L NaCl, 2 mmol/L NaCl, 2 mmol/L EDTA, 1% NP-40) containing a protease inhibitor cocktail (Beyotime, 1 mmol/L). Then, 80 μL of the cell lysates was collected for immunoblotting, and 500 μL of the cell lysates was collected for Co-IP. The lysate was precleared by adding 20 μL protein A/G agarose beads (Beyotime) and incubating it on a rotator at 4 °C for 2 h; then, the beads were spun down at 3000 rpm. The supernatants were collected in a new tube and Anti-Myc-tag mouse mAb (MBL, M192-3) was added. Then, the mixture was incubated on a rotator at 4 °C overnight, and 50 μL beads were added for another 3 h of incubation. The mixture was spun down at 3000 rpm and was washed with 1 mL NP-40 lysis buffer 5 times. The precipitate was collected and 50 μL 2 × loading buffer was added to end the reaction. Finally, Western blotting was used to detect the interaction between the two proteins. The Co-IP ubiquitination assay followed the same protocol, except adding MG-132, anti-DDDDK-tag mouse mAb (MBL, M185-3L), or anti-HA-tag mouse mAb (MBL, M180-3) was used for immunoprecipitation instead.

### 2.7. Lactate Dehydrogenase (LDH) Release Assay

The THP-1 cells were planted in 96-well plates and stimulated as indicated. The supernatants were collected for lactate dehydrogenase (LDH) detection to assess LDH release using a CytoTox 96 Non-Radioactive Cytotoxicity Assay kit (Promega).

### 2.8. ASC Oligomerization

The THP-1 macrophages were primed using LPS (1 μg/mL) for 3 h followed by 3 h of nigericin (10 μmol/L) stimulation. The ASC oligomerization assays were performed as previously described [23]. Only one step was changed. The cells were washed using cold PBS and resuspended in an ice-cold buffer (20 mmol/L HEPES-KOH, pH 7.5, 150 mmol/L KCl, 5 mmol/L EDTA, and protease inhibitor) and lysed by being passed 20 times through a 21-G needle. The lysates were centrifuged at 900× *g* for 8 min to remove the nuclei and unlysed cells. Then, 80 μL of the supernatants was collected for Western blotting and the remaining part of the supernatants was centrifuged at 6200× *g* for 8 min. The pellet fractions were washed twice using PBS and then cross-linked with 2 mmol/L fresh disuccinimidyl suberate (DSS) (Thermo Fisher Scientific, A39267) for 1 h at 37 °C and dissolved in the SDS loading buffer to prevent cross-linking. The cross-linked pellets were separated using SDS–PAGE, and immunoblotting was performed.

### 2.9. IL-1β Estimation

The IL-1β levels were measured using an ELISA kit (1110122, Dakewe, Shenzhen, China) following the manufacturer’s instructions.

### 2.10. Dual-Luciferase Reporter Assay

In brief, 0.5 μg, 0.7 μg, and 0.9 μg of V5-SARS-CoV-2 PLP-TM or SARS-CoV PLP was co-transfected using Flag-RIG-IN, the NF-κB luciferase reporter gene, and TK-Luc into the HEK-293T cells. After 24 h, the relative luciferase was detected using a Dual-Luciferase Reporter Assay System (Promega, E1910).

### 2.11. Immunofluorescence

The THP-1 cells were differentiated into macrophages via treatment with 20 ng/mL PMA for 36 h. The THP-1 macrophages were then cultured using RPMI-1640 or stimulated with 1 μg/mL LPS for 3 h, followed by 10 μM nigericin for 1.5 h. After removing the supernatant and washing it with PBS 3 times, the cells were fixed using 4% polyformaldehyde for 20 min. After washing them with PBS 5 times, the cells were incubated with 0.2% Triton X100 for 20 min, and then washed with PBS 3 times. The cells were incubated in 5% BSA for 1 h and washed with PBS 5 times, and then incubated with the indicated primary antibodies overnight. After washing them with PBS 5 times, the cells were incubated using Alexa Fluor^®^ 594-labeled goat anti-rabbit IgG (H + L) (affinity-purified) (ZSGB-BIO, Beijing, China, ZF-0516) or Alexa Fluor^®^ 594 goat anti-mouse IgG (H + L) (affinity-purified) (ZSGB-BIO, ZF-0513), Alexa Fluor^®^ 488 polyclonal goat anti-rabbit IgG (H + L) (ZSGB-BIO, ZF-0511), and Cy5 conjugated goat anti-mouse IgG (H + L) (Servicebio, Wuhan, China, GB27301). Then, after washing them with PBS for 3 times, the cells were incubated with DAPI for 5 min. Finally, the cells were covered with an anti-quenching agent. The images were acquired using a confocal microscope (Dragonfly, Andor, Belfast, UK). Image processing was conducted using Imaris Viewer 9.8.0.

### 2.12. Statistical Analysis

The sample size was not predetermined using any statistical methods. The experiments were independently repeated at least three times to achieve statistical significance. Data are shown as the means ± SD. The data were analyzed by using the Wilcoxon rank-sum test or two-tailed Student’s *t*-tests, if not otherwise specified. GraphPad Prism 9.0 was used to analyze the data. All of the boxplots in Figure 1 were plotted using “geom_boxplot” in the ggplot2 (ggplot2: Elegant Graphics for Data Analysis. Springer-Verlag New York, 2016) R package. The horizontal line within each box represents the median, and the bottom and top of each box indicate the 25th and 75th percentile. The Wilcoxon rank-sum test was applied to test the significance of the difference between conditions using the “geom_signif” function in the ggsignif (Significance Brackets for ‘ggplot2’) R package. *p*-values < 0.05 were deemed significant (* *p* < 0.05; ** *p* < 0.01; and *** *p* < 0.001); NS means non-significant.

## 3. Results

### 3.1. The Inflammasome Pathway Is Downregulated in the Innate Immune Cells of COVID-19 Patients

Previous studies have shown that, compared with healthy donors, the IL-1β levels in serum from COVID-19 patients were not elevated [24]. To explore this phenomenon in greater depth, we reanalyzed the single-cell RNA sequencing data (accession code HRA000150). Peripheral blood mononuclear cells (PBMCs) from 5 healthy donors, 11 COVID-19 patients (among whom 7 were moderately affected patients and 4 severely affected patients), and 6 convalescent patients were collected for single-cell transcriptional sequencing. When analyzed, eight genes—*NLRP3*, NLR family pyrin-domain-containing 6 (*NLRP6*), PYD-and-CARD-domain-containing protein (*PYCARD*), *IL-1β*, *IL-18*, tripartite-motif-containing 31 (*TRIM31*), F-box and leucine-rich repeat protein 2 (*FBXL2*), and membrane-associated ring-CH-type finger 7 (*MARCH7*)—were used to define the inflammasome pathway score among the natural killer cells, CD14^+^ monocytes, CD16^+^ monocytes, mononuclear dendritic cells, and dendritic cells. Compared with healthy donors and convalescent patients, the inflammasome pathway score was downregulated among the CD14^+^ mononuclear cells, mononuclear dendritic cells, and dendritic cells of the COVID-19 patients (Figure 1A). Furthermore, the expression levels of *NLRP3*, *NLRP6*, *IL-1β*, and *MARCH7* were decreased in COVID-19 patients, while they rose again in convalescent patients (Figure 1B). These results indicated that the NLRP3-associated inflammasome pathway was downregulated in the *SARS-CoV-2*-infected innate immune cells.

### 3.2. SARS-CoV-2 PLP Inhibits the Activation of NLRP3 Inflammasomes and Caspase-1-Mediated GSDMD Cleavage

As we observed that the expression levels of *NLRP3/IL-1β* were downregulated in the innate immune cells from COVID-19 patients, we wanted to further explore the protein encoded by the *SARS-CoV-2* genome involved in the downregulated NLRP3 inflammasome pathway. A previous study has proven that *SARS-CoV-2* PLP negatively regulates IFN-mediated innate immunity [11]. Our group has already reported the negative regulation of IFNβ by other coronavirus PLPs [4,6,7]. In this study, we found that *SARS-CoV-2* PLP-TM could inhibit the activity of the NF-κB pathway (Figure 2), so we suspected that the *SARS-CoV-2* PLP would inhibit the NLRP3 inflammasome to a certain extent. According to previous reports, the expression of PLP, including the downstream transmembrane (TM) domains, is required for processing at the cleavage sites in polyprotein 1a (1ab) [25,26]. Therefore, we generated THP-1 cells stably expressing Flag-tagged *SARS-CoV-2* PLP-TM (aa 1564–aa 2394 of pp1ab). After stimulation via lipopolysaccharides (LPS) and nigericin or ATP, the expression levels of IL-1β (p17) in the cell lysates and supernatants were compared between the Flag-tagged *SARS-CoV-2* PLP-TM-overexpressing cells and the control cells. The results show that the secretion of IL-1β (p17) in the PLP-TM-overexpressing cells was less than in the control cells after stimulation with LPS+ nigericin (Figure 3A,B), as well as with LPS + ATP (Appendix A). Moreover, we examined the expression of the NLRP3-inflammasome-related proteins. Among these proteins, cleaved caspase-1 (p20) was also inhibited by *SARS-CoV-2* PLP-TM after stimulation (Figure 3B). We constructed *SARS-CoV-2* PLP-TM into a V5/HisB-tagged plasmid (Appendix A). Then the plasmids encoding NLRP3, ASC, pro-caspase-1, pro-IL-1β, and V5/HisB-tagged *SARS-CoV-2* PLP-TM were co-transfected into the HEK-293T cells at the same time. The results showed that the secretion levels of IL-1β (p17) in the cell lysates were decreased by *SARS-CoV-2* PLP-TM (Figure 3C). In addition, it seemed that IL-1β (p17) was decreased in the BEAS-2B cells too (Figure 3D). Taken together, these results indicate that *SARS-CoV-2* PLP-TM inhibits the activation of the NLRP3 inflammasome and the secretion of IL-1β.

The activation of the NLRP3 inflammasome recruits pro-caspase-1 and induces pro-caspase-1 to self-cleave. Then, the cleaved caspase-1 (p20) cuts GSDMD into two forms: GSDMD-N and GSDMD-C. GSDMD-N fuses into the membrane, facilitates the formation of membrane pores, and eventually causes cell swelling, which is thought to be the signature of cell pyroptosis [27]. Based on these findings, we deduced that *SARS-CoV-2* PLP-TM inhibited the expression levels of cleaved caspase-1 (p20) (Figure 3B). As a result, we further observed the effect of *SARS-CoV-2* PLP-TM on cell pyroptosis. LPS and nigericin were used to induce THP-1 pyroptosis, and then lactate dehydrogenase (LDH) release assay was conducted to reflect the cell cytotoxicity [28]. Compared with the control cells, the *SARS-CoV-2* PLP-TM-overexpressing THP-1 macrophages showed less LDH release (Figure 3F). Equally, the expression levels of GSDMD-N seemed to decrease in the PLP-TM-overexpressing THP-1 macrophages (Figure 3E). Together, these findings reflect another result of the inhibitory effect of *SARS-CoV-2* PLP-TM in that it inhibited GSDMD cleavage and cell pyroptosis in the THP-1 macrophages. Considering that pyroptosis plays an important role in many immune responses, the results also remind us of the importance of the immune-regulating effect of *SARS-CoV-2* PLP-TM.

### 3.3. SARS-CoV-2 PLP-TM May Coimmunopreciptate with ASC and Blunt the ASC–Caspase-1 Complex

During the process of NLRP3 inflammasome activation, the formation of the NLRP3–ASC–caspase-1 complex is the key to inducing the maturation of IL-1β and the formation of GSDMD-N [1]. The abovementioned results indicate that *SARS-CoV-2* PLP-TM might inhibit the activation of the NLRP3 inflammasome and pyroptosis. However, we are still unsure about how *SARS-CoV-2* PLP-TM reduces the activation of the NLRP3 inflammasome. Therefore, we detected the interaction between *SARS-CoV-2* PLP-TM and NLRP3/ASC/caspase-1/pro-IL-1β separately via the co-IP assay. As a result, we did not observe the interaction between *SARS-CoV-2* PLP-TM and NLRP3, pro-caspase-1, or pro-IL-1β (Appendix A); however, we observed that *SARS-CoV-2* PLP-TM could be coimmunoprecipitated using ASC (Figure 4A, lane4, Figure 4B). To further clarify whether PLP would interrupt the formation of the ASC–caspase-1 complex, we co-transfected the indicated plasmids into the HEK-293T cells and found that PLP could interrupt the formation of the ASC–caspase-1 complex (Figure 4C, column 3). All these results indicate that *SARS-CoV-2* PLP-TM may coimmunoprecipitate with ASC and weaken the formation of the ASC–caspase-1 complex so that it can affect the activation of the NLRP3 inflammasome and pyroptosis.

### 3.4. SARS-CoV-2 PLP Reduces the Oligomerization of ASC and ASC Speck Formation

The oligomerization of ASC is an important sign in the activation of the NLRP3 inflammasome. The oligomerization of ASC results in the formation of ASC specks, which act as a signal amplification mechanism for the inflammasomes [29]. Meanwhile, the formation of ASC oligomers recruits pro-caspase-1 and induces pro-caspase-1 cleavage [30]. As shown in Figure 5, the ASC was redistributed from the nucleus to the cytoplasm and organized into specks after stimulation using LPS + nig. However, there were fewer ASC specks in the THP-1 cells stably expressing *SARS-CoV-2* PLP-TM than in the control cells (Figure 5A). To observe the effect of PLP-TM on ASC oligomers, V5-tagged *SARS-CoV-2* PLP-TM plasmids were co-transfected with plasmids encoding NLRP3, ASC, and pro-caspase-1 into the HEK-293T cells for 24 h; then, the ASC oligomers were detected using immunoblotting according to previous reports [23,31]. It was shown that PLP-TM could decrease the oligomerization of ASC in the HEK-293T cells (Figure 5B). *SARS-CoV-2* PLP-TM could reduce the ASC oligomers stimulated by LPS and nig in the THP-1 macrophages (Figure 5C). Furthermore, similar results were observed in the BEAS-2B cells (Appendix A). These results indicate that *SARS-CoV-2* PLP-TM might decrease the oligomerization of ASC and ASC specks when the NLRP3 inflammasome pathway is excessively activated.

### 3.5. SARS-CoV-2 PLP Reduces the Ubiquitination of ASC

Post-translational modification plays an important role in the activation of the NLRP3 inflammasome. Among these modifications, ubiquitination and deubiquitination are of great significance. The deubiquitination of NLRP3 and the ubiquitination of ASC, pro-caspase-1, and pro-IL-1β are required for the activation of the NLRP3 inflammasome [14,32]. In addition, *SARS-CoV-2* PLP-TM has a Ub-binding domain and could play a role in the deubiquitination of protein substrates [33]. As a result, we investigated whether *SARS-CoV-2* could mediate the deubiquitination of NLRP3, ASC, and pro-caspase-1. Plasmids encoding *SARS-CoV-2* PLP-TM were transfected into the HEK-293T cells together with plasmids encoding NLRP3 or ASC or pro-caspase-1 and ubiquitin.

The ubiquitination of NLRP3, ASC, and pro-caspase-1 was observed in the THP-1 cells (Figure 6A–C, column 2). When treated with *SARS-CoV-2* PLP-TM, the deubiquitinating effect of PLP-TM on NLRP3 and pro-caspase-1 seemed not to be significant (Figure 6A,C, lane3). However, the reduction in the ubiquitination of ASC was obvious (Figure 6B, lane3). This result indicates that *SARS-CoV-2* PLP-TM might inhibit the activation of the NLRP3 inflammasome via the deubiquitination of ASC, considering that the ubiquitination of ASC contributes to the full function of the NLRP3 inflammasome.

Among the reported types of ubiquitination, lysine(K)48- and K63-linked ubiquitin are the fully characterized types. To further determine the effects of *SARS-CoV-2* PLP-TM-mediated deubiquitination on ASC, we co-transfected V5-tagged PLP-TM, Flag-ASC, and HA-tagged K48- or K63-linked ubiquitin into the HEK-293T cells. We observed that *SARS-CoV-2* PLP-TM could reduce both the K48- and K63-linked ubiquitination of ASC (Figure 6D,E, column 3). These results reflect the possibility that *SARS-CoV-2* PLP-TM might reduce the K48- and K63-linked ubiquitination of ASC without selectivity for substrates.

## 4. Discussion

In this study, we first found that the gene expression of NLRP3 and IL-1β was downregulated in the CD14^+^ and CD16^+^ monocytes from COVID-19 patients, while MARCH7 and IL-18 declined only in the CD16^+^ monocytes. Further studies showed that *SARS-CoV-2* PLP-TM might interrupt the formation of the ASC–caspase-1 complex, inhibit the activation of caspase-1, and subsequently block IL-1β maturation. As activated caspase-1 (p20) cleaves GSDMD and mediates pyroptosis, PLP-TM inhibits cell pyroptosis and the release of IL-1β. Specifically, PLP-TM interacts with ASC and reduces the K48- and K63-linked ubiquitination of ASC, which decreases the oligomerization of ASC and ASC specks (Figure 7). The finding that PLP-TM encoded by the *SARS-CoV-2* genome interacts with host factors to antagonize antiviral immunity might explain why asymptomatic patients accounted for one-third of the total number of infected patients [34].

Like other coronaviruses, the *SARS-CoV-2* genome first causes translation and encodes polyprotein pp1a(1ab) containing NSP1-16 and the expression of other structural proteins. At the same time, PLP cleaves NSP1-3 and regulates virus replication [35,36]. Previous research has shown that SARS, NL63, and PEDV PLPs could inhibit IFNβ innate immunity [5], while *SARS-CoV-2* NSP1 and NSP13 suppress caspase-1 and NLRP3 inflammasome activation [37]. Our findings in this research showed that *SARS-CoV-2* PLP also negatively regulated the NLRP3 inflammasome pathway, inhibited cell pyroptosis, and finally reduced IL-1β production and secretion. These findings reflect that PLP functions as an innate immunity antagonist and plays a novel role in viral immune escape, which may repress viral clearance and be beneficial for virus replication.

On the other hand, other viral proteins, such as N protein, S protein, and so on, show a more significant cumulative effect of activation on immunity after the virus’s large-scale replication and release [38], which may help to explain why the level of inflammasome cytokines during the late stages of infection is not consistent with the early stages. For example, inflammatory cytokines such as IL-1β are not significantly increased within 96 h after infection and show no significant increase in the serum from asymptomatic and mild patients in the early stages of the disease, but they are significantly increased in the late stage of infection [16,39].

The NLRP3 inflammasome is a common anti-virus immune pathway that responds to viral invasion [40]. Upon activation, NLRP3 can assemble with ASC, and the complex finally activates caspase-1. ASC is an adaptor protein containing two domains, pyrin and CARD, which link upstream NLRP3 with downstream pro-caspase-1 [41]. The formation of ASC dimers is the key to constituting ASC specks [42], and the formation of ASC specks is indicative of NLRP3 inflammasome activation, recruiting pro-caspase-1 to self-catalyze. The activated form of caspase-1 (p20) induces IL-1β maturation and cleaves the full length of GSDMD into GSDMD-N and GSDMD-C. GSDMD-N forms pores on the membrane that release IL-1β, and finally, the cell swells to death [27,41]. In this research, we found that the interaction between *SARS-CoV-2* PLP-TM and ASC might block the interaction between ASC and pro-caspase-1 so that the cleaved caspase-1 (p20) is suppressed. We also observed that *SARS-CoV-2* reduced ASC specks and restrained the oligomerization of ASC.

Furthermore, post-translational modifications, including ubiquitination, could regulate immune responses, such as IKK activation, mitogen-activated protein kinase (MAPK) activation, TBK1/IKKε activation, NLR signaling, and so on [43]. As reported previously, ubiquitination is also essential for the oligomerization of ASC [14,32], and the ubiquitination of ASC contributes to the fully functional activation of the NLRP3 inflammasome [44]. In our study, *SARS-CoV-2* PLP-TM interacted with ASC and performed deubiquitinating (DUB) activities, contributing to the inhibition of ASC oligomerization, and preventing the formation of NLRP3 complexes. Different types of ubiquitination are thought to modulate the activities of inflammasome’s innate immune signaling. Here, we observed that *SARS-CoV-2* PLP-TM reduces the K48- and K63-linked ubiquitination of ASC without selectivity for substrates. Future experiments will focus on how the sequence changes in PLP can influence the ubiquitination and activation of the ASC and NLRP3 inflammasome pathways.

Meanwhile, *SARS-CoV-2* PLP-TM suppresses the activation of pro-caspase-1, which in turn decreases the cleavage of GSDMD and weakens cell pyroptosis. Then, the release of IL-1β is also inhibited. According to previous reports, several CoV PLPs negatively regulated the NF-κB pathway [45], while the activated NF-κB pathway may upregulate the NLRP3 inflammasome response [46]. In this research, we also found that *SARS-CoV-2* PLP-TM attenuated the activation of the NF-κB pathway. All of these findings may indicate that *SARS-CoV-2* PLP-TM negatively regulates the NLRP3 inflammasome response from the upstream pathway. Further studies are needed to determine the relationship between the inhibitory effect of PLP and other activities.

In conclusion, we demonstrated the novel role of *SARS-CoV-2* PLP-TM in host innate immune regulation. The expression levels of NLRP3, NLRP6, IL-1β, and MARCH7 were decreased in COVID-19 patients, and this reflected the inhibition of the NLRP3 inflammasome pathway. SARS-CoV-2 PLP reduces ASC specks, ASC oligomerization, and the ubiquitination of ASC, which subsequently inhibits the production and secretion of IL-1β and negatively regulates the NLRP3 inflammasome pathway and cell pyroptosis. As a result, after infection with *SARS-CoV-2* and the encoding of PLP, the host cannot produce effective antiviral immunity. This status helps the virus to evade antiviral immunity and is beneficial to maintaining the immune balance and promoting virus replication. These findings give us a better understanding of the flex function of CoV encoding proteases and provide a new perspective on host innate immune responses against *SARS-CoV-2* and other viruses.

## Figures and Tables

**Figure 1 microorganisms-11-02799-f001:**
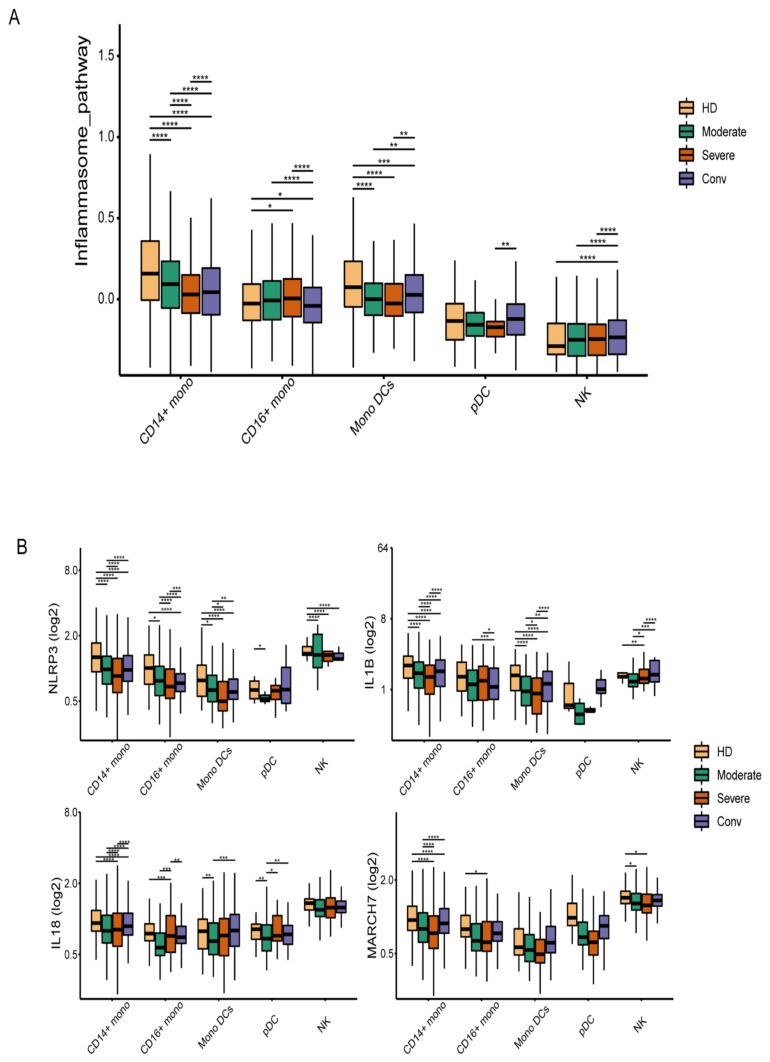
Analysis of the NLRP3 inflammasome pathway based on single-cell transcriptional sequence data in innate immune cells from COVID-19 patients. (**A**) The expression level of the inflammasome pathway (*NLRP3*, *NLRP6*, *PYCARD*, *IL-1β*, *IL-18*, *TRIM31*, *FBXL2*, *MARCH7*) across four conditions. (**B**) The expression levels of *NLRP3*, *MARCH7*, *IL-1β*, and *IL-18* in innate immune cells from each group were calculated. The single-cell transcriptional profiling of innate immune cells was obtained from healthy donors (*n* = 5), patients moderately infected with *SARS-CoV-2* (*n* = 7), severely affected patients (*n* = 4), and convalescent patients (*n* = 6). The Wilcoxon rank-sum test was used to test the significance of the difference between conditions. * *p* < 0.05, ** *p* < 0.01, *** *p* < 0.001, **** *p* < 0.0001. (HD: Healthy donors; Conv: Convalescent; *NLRP3*: NLR family pyrin-domain-containing-3; NLRP6: NLR family pyrin-domain-containing-6; *ASC/PYCARD*: apoptosis-associated speck-like protein containing a caspase recruitment domain, also known as *PYCARD* (PYD-and-CARD-domain-containing protein); *IL-1β*: Interleukin-1 β; *IL-18*: Interleukin-18; *TRIM31*: Tripartite-motif-containing-31; *FBXL2*: F-box and leucine-rich repeat protein 2; *MARCH7*: Membrane-associated ring-CH-type finger 7).

**Figure 2 microorganisms-11-02799-f002:**
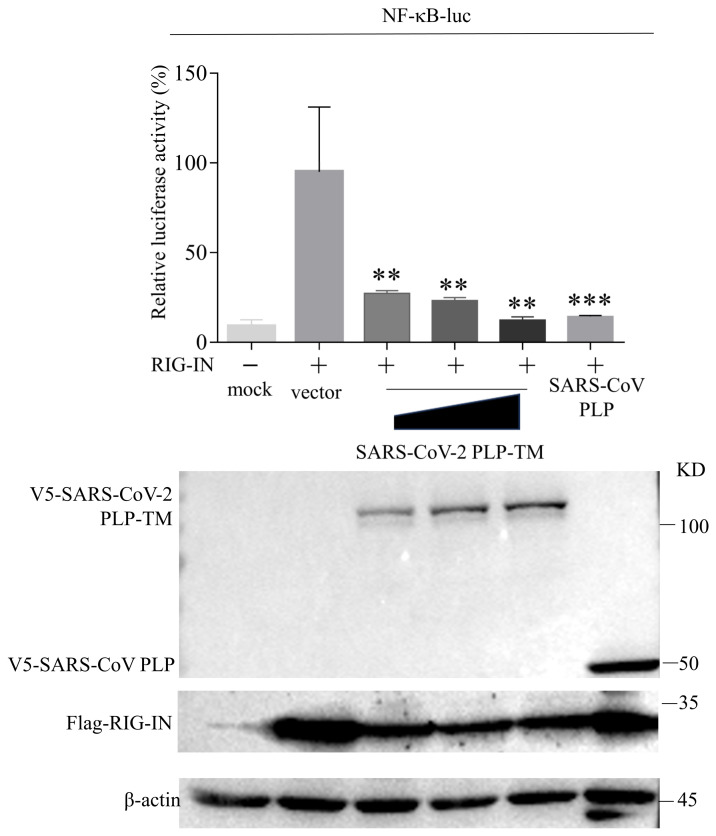
*SARS-CoV-2* papain-like protease (PLP) inhibits the promotor activity of nuclear factor kappa-B (NF-κB). 0.5 μg, 0.7 μg, 0.9 μg V5-*SARS-CoV-2* PLP-TM or *SARS-CoV* PLP was co-transfected with Flag-RIG-IN, NF-κB luciferase reporter gene, and TK-Luc into HEK-293T cells, as detected via a dual-luciferase reporter assay. *SARS-CoV* PLP was transfected as a positive control. ** *p* < 0.01; *** *p* < 0.001.

**Figure 3 microorganisms-11-02799-f003:**
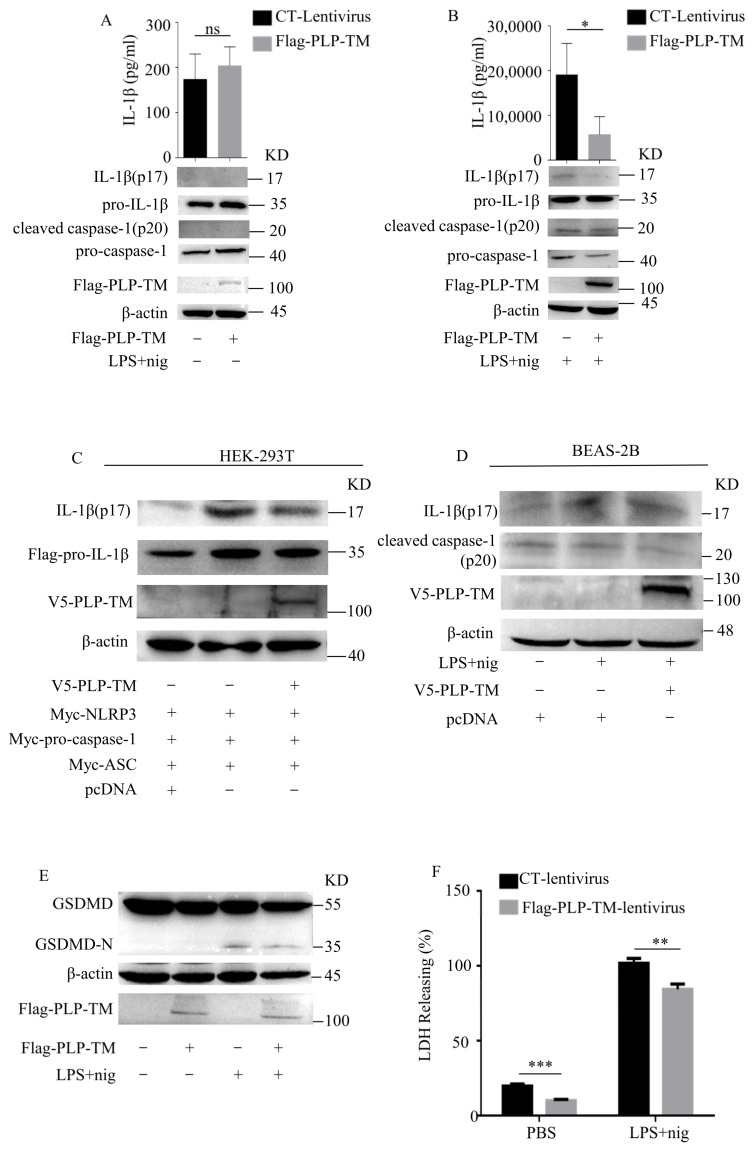
*SARS-CoV-2* PLP inhibits caspase-1 cleavage and the maturation of IL-1β. (**A**) THP-1 cells stably expressing Flag-tagged *SARS-CoV-2* PLP-TM or the control vector were differentiated into macrophages via treatment with 20 ng/mL PMA for 36 h and then incubated with PBS. Supernatants were collected for ELISA assay to determine IL-1β levels, and cell lysates used for Western blotting. (**B**) THP-1 cells stably expressing Flag-tagged *SARS-CoV-2* PLP-TM or the control vector were differentiated into macrophages via treatment with 20 ng/mL PMA for 36 h and then pretreated with 1 μg/mL LPS for 3 h, followed by 10 μmol/L nigericin for 1.5 h. The supernatants were collected for ELISA assay to determine IL-1β levels and cell lysates used for Western blotting to analyze IL-1β(p17), pro-IL-1β, caspase-1 and cleaved caspase-1 (p20). (**C**) HEK-293T cells were co-transfected with Myc-tagged NLRP3, pro-caspase-1, ASC, and Flag-tagged pro-IL-1β for 24 h. The cell lysates were collected for Western blotting to analyze IL-1β(p17) and Flag-pro-IL-1β. (**D**) BEAS-2B cells were transfected using V5-vector or V5-*SARS-CoV-2* PLP-TM for 24 h and stimulated with 1 μg/mL LPS for 12 h plus 10 μmol/L nigericin for 3 h. The cell lysates were collected for Western blotting to analyze IL-1β(p17) and cleaved caspase-1 (p20). (**E**) THP-1 cells stably expressing Flag-tagged *SARS-CoV-2* PLP-TM or the control vector were differentiated into macrophages via treatment with 20 ng/mL PMA for 36 h and then pretreated using PBS or 1 μg/mL LPS for 3 h, followed by 10 μmol/L nigericin for 1.5 h. Th cell lysates were collected for Western blotting to analyze the expression of GSDMD and GSDMD-N. (**F**) THP-1 cells stably expressing Flag-tagged *SARS-CoV-2* PLP-TM or the control vector were differentiated into macrophages via treatment with 20 ng/mL PMA for 36 h and then pretreated with PBS or 1 μg/mL LPS for, 3 h followed by 10 μmol/L nigericin for 1.5 h. LDH release was examined using the lactate dehydrogenase (LDH) release assay in supernatants. NS, non-significant; * *p* < 0.05; ** *p* < 0.01; *** *p* < 0.001. (LPS: lipopolysaccharide; GSDMD: gasdermin D, a member of the gasdermin family; GSDMD-N: the N-terminal of GSDMD; LDH: lactate dehydrogenase).

**Figure 4 microorganisms-11-02799-f004:**
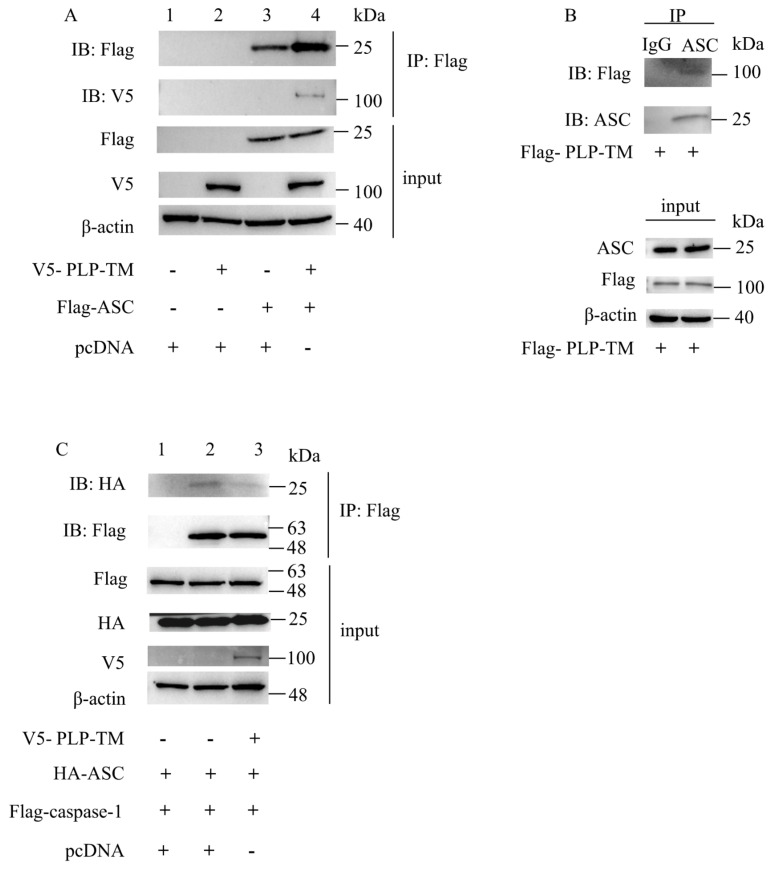
*SARS-CoV-2* PLP interacts with ASC and interrupts the formation of the ASC–caspase-1 complex. (**A**) HEK-293T cells were transfected with plasmids encoding Myc-tagged ASC (columns 3 and 4), or with plasmids encoding V5-tagged SARS-CoV-2 PLP-TM (columns 2 and 4) and with an empty V5-tagged vector. Cell lysates were collected and immunoprecipitated using antibodies against Myc, followed by immunodetection with antibodies against V5. An immunoblot analysis of cell lysates was conducted as indicated. (**B**) PMA-differentiated THP-1 macrophages were stably infected using Flag-*SARS-CoV-2* PLP-TM-Lentivirus and were incubated with 1 μg/mL LPS for 3 h plus 10 μmol/L Nigericin for 1.5 h. Cell lysates were collected and immunoprecipitated with antibodies against ASC, followed by immunodetection with antibodies against Flag. An immunoblot analysis of the cell lysates was conducted as indicated. (**C**) Flag-pro-caspase-1+HA-ASC was transfected with V5-*SARS-CoV-2* or pcDNA into the HEK-293T. After 24 h, it was immunoprecipitated using antibodies against IgG or Flag, followed by immunodetection using antibodies against HA. An immunoblot analysis of the cell lysates was conducted as indicated.

**Figure 5 microorganisms-11-02799-f005:**
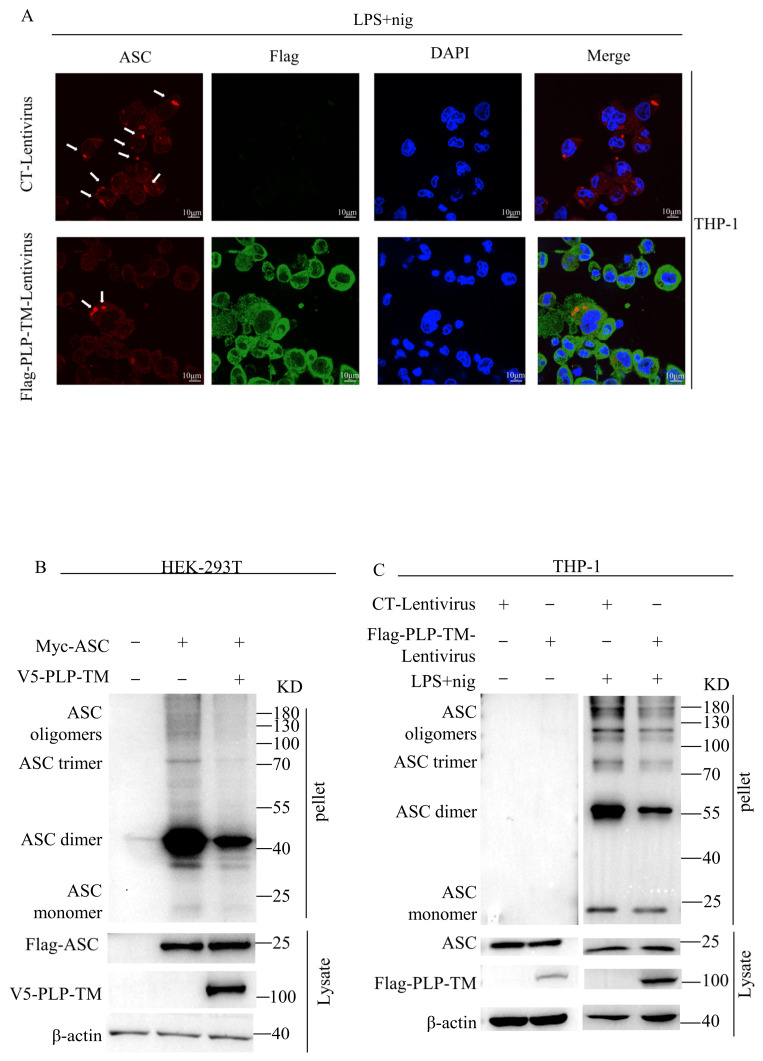
*SARS-CoV-2* PLP reduces the oligomerization of ASC and ASC specks. (**A**) PMA-differentiated THP-1 macrophages were stably infected with CT-Lentivirus or Flag-*SARS-CoV-2* PLP-TM-Lentivirus and were incubated with 1 μg/mL LPS for 3 h plus 10 μmol/L nigericin for 1.5 h. The sub-cellular locations of ASC (red), Flag-*SARS-CoV-2* PLP-TM (green), and the nucleus marker (blue) were visualized via confocal microscopy. The ASC specks are indicated by white arrows. The objects were 100×-magnified under an oil microscope. Scale bar: 10 μm. (**B**) HEK-293 T cells were co-transfected using Myc-tagged NLRP3, Myc-pro-caspase-1, Flag-ASC, and Flag-tagged pro-IL-1β. Cell lysates were collected for the immunoblotting analysis of ASC oligomerization. (**C**) THP-1 macrophages stably expressing Flag-tagged *SARS-CoV-2* PLP-TM or the control vector were pretreated with 1 μg/mL LPS, followed by 10 μmol/L nigericin for 1.5 h. Cell lysates were collected for the immunoblotting analysis of ASC oligomerization.

**Figure 6 microorganisms-11-02799-f006:**
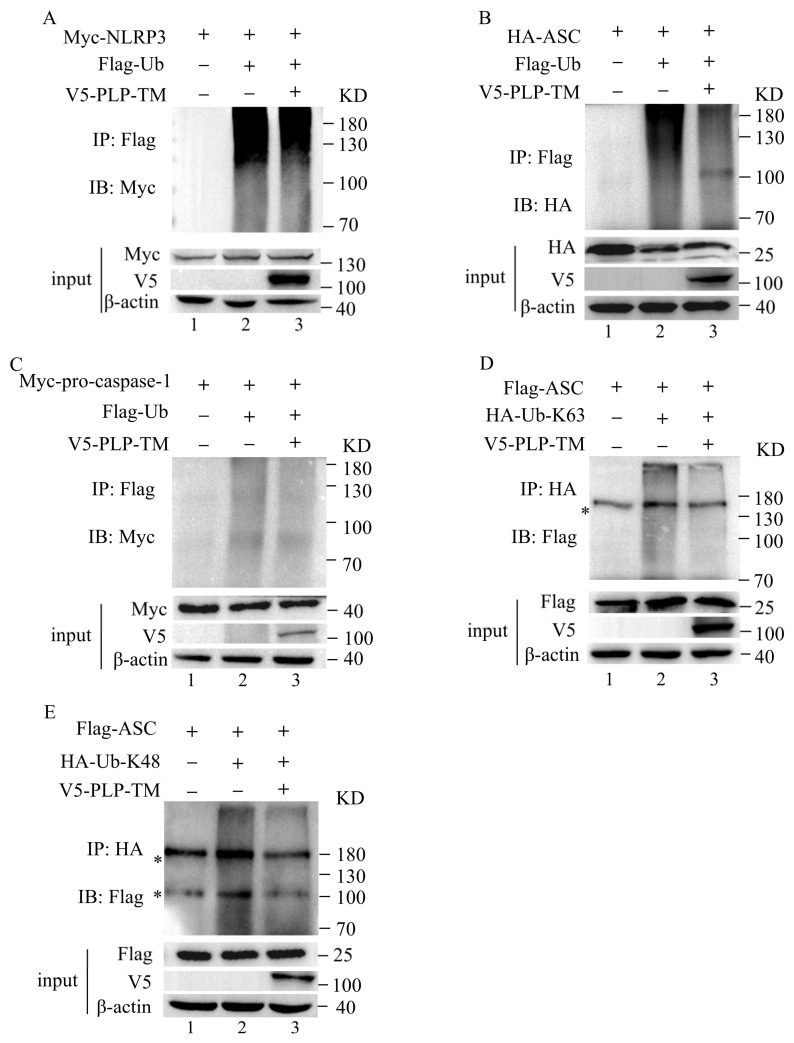
*SARS-CoV-2* PLP decreases the ubiquitination of ASC. (**A**,**B**) Immunoblot analysis of lysates from HEK-293T cells transfected with V5-tagged *SARS-CoV-2* PLP-TM, Flag-tagged ubiquitin (Flag-Ub), and Myc-tagged NLRP3 (**A**) or HA-tagged ASC (**B**) or Myc-tagged pro-caspase-1 (**C**), followed by immunoprecipitation with antibodies against Flag and immunodetection with antibodies against Myc or HA. The immunoblotting of the cell lysates was detected as indicated. (**D**,**E**) HEK-293T cells transfected with Flag-tagged ASC, V5-tagged *SARS-CoV-2* PLP-TM, and HA-tagged K63- or HA-tagged K48-linked ubiquitin (the most fully characterized types of ubiquitination). The immunoprecipitation and immunoblotting of cell lysates were detected as indicated. * Non-specific band.

**Figure 7 microorganisms-11-02799-f007:**
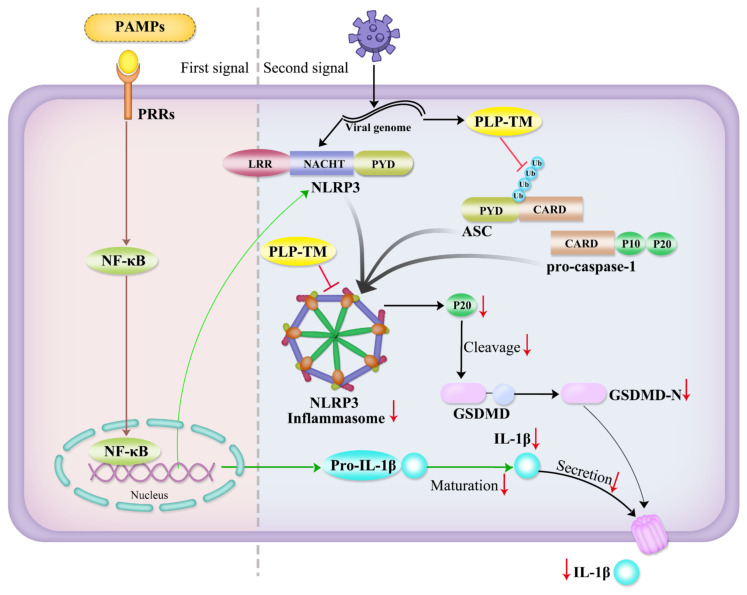
A hypothetical model describing the mechanisms by which *SARS-CoV-2* PLP negatively regulates the NLRP3 inflammasome immune pathway. *SARS-CoV-2* PLP-TM interacts with ASC and reduces the K48- and K63-linked ubiquitination of ASC, which decreases the oligomerization of ASC. As a result, *SARS-CoV-2* PLP-TM might block IL-1β maturation, and inhibit the activation of GSDMD. Finally, the release of IL-1β is reduced.

## Data Availability

The data underlying Figure 1 and Appendix A are openly available in the Genome Sequence Archive of the Beijing Institute of Genomics (BIG) Data Center at https://ngdc.cncb.ac.cn/gsa-human/browse/HRA000150. (accessed 11 October 2022)

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
