# Peer review of "SARS-CoV-2 Papain-like Protease Negatively Regulates the NLRP3 Inflammasome Pathway and Pyroptosis by Reducing the Oligomerization and Ubiquitination of ASC"

_microorganisms, 2023, doi:10.3390/microorganisms11112799_

Round 1
Reviewer 1 Report
Comments and Suggestions for Authors
Meng et al. investigated the molecular function of papain-like protease (PLP)of SARS-CoV-2 on the NLRP3 inflammasome pathway, and successfully found that the expression of inflammasome-related genes was down-regulated in monocytes of COVID-19 patients. The authors also found that SARS-CoV-2 PLP interfered the interaction between ASC and caspase-1 by directly biding to ASC. Furthermore, the authors observed that PLP modified the ubiquitination state of the components of inflammasome, possibly associating with the negative regulation of the inflammasome function.
The manuscript is theoretical and the data are enough to show the authors’ conclusion. This is a report of high-quality molecular assays and will attract considerable attention from readers. However, the reviewer would like to suggest several minor concerns that might be addressed by the authors before publication.
Minor concerns:
1. Lines 133-136. In Materials and Methods 2.4., sentences should be re-checked to show Methods clearly and to follow correct grammar.
2. Figure 1. Please indicate what the abbreviation “HD” and “Conv” mean in the legend.
3. Figure 1A. Please indicate the unit of the Y-axis.
4. Figure 3D and 3E. The reviewer wonders if the amounts of IL-1b and GSDMD-N really decreased. If the authors quantified the WB images, please show the quantified results. If not, the reviewer would like to suggest to modify the word “decreased” to milder expression (using “it seemed” or “likely”, etc.) in the main text.
Author Response
Response to Reviewer 1 Comments
|
||
1. Summary |
|
|
Thank you very much for taking the time to review this manuscript. Please find the detailed responses below and the corresponding revisions/corrections highlighted/in track changes in the re-submitted files.
|
||
2. Questions for General Evaluation |
Reviewer’s Evaluation |
Response and Revisions |
Does the introduction provide sufficient background and include all relevant references? |
Ye |
Thank you for your item-by-item evaluation. |
Are all the cited references relevant to the research? |
Yes |
|
Is the research design appropriate? |
Yes |
|
Are the methods adequately described? |
Yes |
|
Are the results clearly presented? |
Ye |
|
Are the conclusions supported by the results? |
Yes |
|
3. Point-by-point response to Comments and Suggestions for Authors |
||
Comments 1: Lines 133-136. In Materials and Methods 2.4., sentences should be re-checked to show Methods clearly and to follow correct grammar. Response 1: We sincerely thank reviewer for the kind remarks. We have corrected the corresponding sentences in lines 140-144. |
||
Comments 2: Figure 1. Please indicate what the abbreviations “HD” and “Conv” mean in the legend. Response: We sincerely thank the reviewer for the kind remarks. “HD” represents “Healthy Donor”, “Conv” represents convalescent. We have pointed out the abbreviations “HD” and “Conv” in the legend of Figure 1.
Comments 3: Figure 1A. Please indicate the unit of the Y-axis. Response: We sincerely thank reviewer for the kind remarks. Figure 1A depicts the module scores for the inflammasome pathway in each cell cluster. The values of the Y-axis were calculated using the “AddModuleScore” function in the Seurat package. They represent the relative expression levels of the inflammasome pathway and therefore have no unit. Comments 4. Figure 3D and 3E. The reviewer wonders if the amounts of IL-1b and GSDMD-N really decreased. If the authors quantified the WB images, please show the quantified results. If not, the reviewer would like to suggest to modify the word “decreased” to milder expression (using “it seemed” or “likely”, etc.) in the main text. Response: The suggestion of reviewer is very meaningful and thanks a lot. According to this suggestion, we have modified the words in a milder expression. The corrected corresponding sentences were in lines 274-275 and lines 287-288 in the revised manuscript. |
||
4. Response to Comments on the Quality of English Language |
||
Point 1: I am not qualified to assess the quality of English in this paper. |
||
Response 1: Thanks for your suggestions on the quality of the English language, we have polished the language according to the weblink the editor provided. All the modifications were highlighted in the revised manuscript. |
Reviewer 2 Report
Comments and Suggestions for Authors
The manuscript reports about important finding of regulation of host innate immunity with the SARS-CoV-2 specific protease.
However, without the necessary decryption of all abbreviations it's hardly possible to understand some significant details. Unfortunately, English grammar also needs extensive editing. Problem points are highlighted in yellow in the attached file.
Introduction.
Сitation method with numbers in regular font without brackets seems to be uncommon and is not the same throughout the manuscript.
Line 30
There are not "nucleotides" in RNA and DNA but nucleotide residues connected with phosphodiester bonds. Comparison of full-length genomes of SARS-CoV-2 shows variations of lengths. Not all of them have 29903 nucleotide residues.
Line 33
What specificity of papain-like protease (PLP)? What activities is known for the enzyme (endopeptidase, aminopeptidase, dipeptidyl peptidase or something else)?
Lines 39-45
There are 3 main types of interferons (IFN). Which one is regulated by coronaviral PLP?
Does human IFN has a cleavage site or a few potential sites specific for PLP?
Lines 61-62
Definition of cytokine storm with elevated levels of IL-6, IL-2, IL-4, IL-10, and TNF-α is missing. It is not the common law for all patients with severe COVID-19.
Line 56
What means interaction of PLP with Apoptosis-associated speck-like protein containing a caspase recruitment domain (ASC)? Proteolytic hydrolysis? What protein fragments were found including both theoretically expected and experimentally observed?
ASC oligomerization and ubiquitination are not described in Introduction.
Materials and methods.
Line 81.
What is RIMP-1640? Probably, rich culture medium RPMI 1640?
Line 122.
Origin of all tissue cultures (HEK-293T cells, BEAS-2B cells and THP-1) should be added.
What stands for P/S? Antibiotics or something else. It's hardly possible to guess.
Lines 133-138.
Construction of the recombinant plasmids is not complete.
Isolation of original SARS-CoV-2 RNA and reverse transcription with subsequent PCR with unknown primers are not described at all.
Why PLP-TM (transmembrane?). Is it a truncated gene and recombinant protein fragment or full-length protein?
Line 140.
Structures of primers specific to human pro-IL-1β also remain unknown.
What promoter(s) control transcription of the cloned genes in the transfected eukaryotic cells? Probably, schemes of cloning and subsequent subcloning would be helpful.
Line 144
Presentation of methods as "5E+8 TU/ml" is not common and is not acceptable.
Line 164
Lactate dehydrogenase (LDH) is a cytoplasmic enzyme found inside all living cells. After the plasma membrane damages LDH is released into the cultural medium. But cytotoxicity may be without leakage of enzyme(s) through pores of cellular membranes. Therefore, other methods based on mitochondrial reductase that is active in the living cells only (MTT, XTT, WST) are widely used to estimate cytotoxicity.
Line 190.
What means "technical replicates"?
Results.
Figure 1. Abbreviations are not determined. Standard deviations shown as vertical black lines do not correspond multiple stars that show "the significance of the difference between conditions. *P<0.05, ** P<0.01, *** P<0.001, **** P<0.0001."
Figure 2 is multipanel. But the legend corresponds to the upper luciferase activity assay. Moreover, even for luciferase activity the significance of 4 right columns marked with "+ SARS-CoV PLP" is unclear. What difference among them? PLP from the SARS-CoV or from SARS-CoV-2?
Figure 3 seems overloaded and without detailed description in the legend is unclear.
Figure 5 is mentioned in the text before figure 4. It must be corrected.
Schemes of co-immunoprecipitation and cross-linking could make clear possible evidences of so called "interaction" of protease with protein and oligomerization. Addional evidence might be electrophoresis in native conditions without previous denaturation of samples.
Figure 5, part A.
Scale bars are without numbers. Results of confocal fluorescent microscopy should be shown with the scale bars and description of magnification in the legend.
What difference between specks and fluorescent micro- or nanoparticles?
Unfortunately, without clear presentation and description of numerous available results at present it's difficult to discuss and to compare with other research.

Comments on the Quality of English Language
Extensive editing of English language required.
Author Response
Response to Reviewer 2 Comments
|
||||
1. Summary |
|
|
||
Thank you very much for taking the time to review this manuscript. Please find the detailed responses below and the corresponding revisions/corrections highlighted/in track changes in the re-submitted files.
|
||||
2. Questions for General Evaluation |
Reviewer’s Evaluation |
Response and Revisions |
||
Does the introduction provide sufficient background and include all relevant references? |
Can be improved |
Thank you for your evaluation. We have improved the corresponding parts according to your suggestions. |
||
Are all the cited references relevant to the research? |
Can be improved |
Thank you for your evaluation. We have improved the corresponding parts according to your suggestions. |
||
Is the research design appropriate? |
Can be improved |
Thank you for your evaluation. We have improved the corresponding parts according to your suggestions. |
||
Are the methods adequately described? |
Must be improved |
Thank you for your evaluation. We have checked the methods and improved them. |
||
Are the results clearly presented? |
Can be improved |
Thank you for your item-by-item evaluation. We have improved the corresponding parts according to your suggestions. |
||
Are the conclusions supported by the results? |
Can be improved |
Thank you for your evaluation. We have improved the corresponding parts according to your suggestions. |
||
3. Point-by-point response to Comments and Suggestions for Authors |
||||
Comments 1: Introduction. Сitation method with numbers in regular font without brackets seems to be uncommon and is not the same throughout the manuscript. Response: We sincerely thank the reviewer for the kind remarks. We have corrected the citation in the revised manuscript. |
||||
Comments 2: Line 30: There are not "nucleotides" in RNA and DNA but nucleotide residues connected with phosphodiester bonds. Comparison of full-length genomes of SARS-CoV-2 shows variations of lengths. Not all of them have 29903 nucleotide residues. Response: We would like to thank the reviewer for the very constructive suggestion. We have corrected the corresponding sentences in lines 30-31 in the revised manuscript. |
||||
Comments 3: Line 33: What specificity of papain-like protease (PLP)? What activities is known for the enzyme (endopeptidase, aminopeptidase, dipeptidyl peptidase or something else)? Response: We sincerely thank the reviewer for the kind remarks. According to the published reports, SARS-CoV-2 PLP exhibits catalytic activity in processing the cleavage of nsp1¯nsp2, nsp2¯nsp3, and nsp3¯nsp4, participates in virus replication in collaboration with other proteases like 3C-like protease (chymotrypsin-like protease, 3CL) and recognizes the C‐terminus of ubiquitin acting as deubiquitinase. We have corrected the descriptions of the papain-like protease in lines 36-37. Comments 4: Lines 39-45: There are 3 main types of interferons (IFN). Which one is regulated by coronaviral PLP? Response: We sincerely thank the reviewer for the kind remarks. According to previous studies, among 3 main types of IFN (IFNα, IFNβ and IFNγ), PLP mainly negatively regulated IFNβ related immune pathways. The corresponding parts have been corrected in lines 40-47, 48-53. Comments 5: Does human IFN has a cleavage site or a few potential sites specific for PLP? Response: We sincerely thank the reviewer for the kind remarks. In a previous study, we observed that PLP negatively regulated the expression of IFNβ and related immune response in the upstream pathway mediated by RIG-I, STING, etc. Moreover, another laboratory has reported that direct cleavage of IRF3 by NSP3/papain-like protease could explain the blunted type-I IFN response seen during SARS-CoV-2 infections (Moustaqil M, Ollivier E, Chiu HP, et al. SARS-CoV-2 proteases PLpro and 3CLpro cleave IRF3 and critical modulators of inflammatory pathways (NLRP12 and TAB1): implications for disease presentation across species. Emerg Microbes Infect. 2021;10(1):178-195. doi:10.1080/22221751.2020.1870414).
Comments 6: Lines 61-62: Definition of cytokine storm with elevated levels of IL-6, IL-2, IL-4, IL-10, and TNF-α is missing. It is not the common law for all patients with severe COVID-19.
Furthermore, other studies have reported that higher levels of cytokine storm are associated with more severe disease development in COVID-19 patients. Among them, IL-6 and IL-10 can be used as predictors for fast diagnosis of patients with a higher risk of disease deterioration (Han H, Ma Q, Li C, et al. Profiling serum cytokines in COVID-19 patients reveals that IL-6 and IL-10 are disease severity predictors. Emerg Microbes Infect. 2020;9(1):1123-1130. doi:10.1080/22221751.2020.1770129). According to the constructive suggestion of the reviewer, we used a milder expression in the revised manuscript as “In most of patients with severe COVID-19, the serum cytokines including IL-6, IL-2, IL-10, and TNF-α were elevated” in lines 66-67. Comments 7: Line 56: What means interaction of PLP with Apoptosis-associated speck-like protein containing a caspase recruitment domain (ASC)? Proteolytic hydrolysis? What protein fragments were found including both theoretically expected and experimentally observed? Response: We sincerely thank the reviewer for the kind remarks. According to our own findings and other reports (Park HS, Liu G, Thulasi Raman SN, Landreth SL, Liu Q, Zhou Y. NS1 Protein of 2009 Pandemic Influenza A Virus Inhibits Porcine NLRP3 Inflammasome-Mediated Interleukin-1 Beta Production by Suppressing ASC Ubiquitination. J Virol. 2018;92(8):e00022-18. Published 2018 Mar 28. doi:10.1128/JVI.00022-18), it may be suggested that the effects resulting from the spatial interaction of the ASC and viral proteins (such as PLP or IAV NS1) might be mainly observed from the function of the downstream signaling pathway. The spatial interaction of PLP with ASC might hinder the formation of the ASC-caspase-1 complex, thus interrupting the NLRP3-ASC-caspase1 complex, which subsequently inhibits the activation of pro-IL-1β (Results 3.2, 3.3). In the revised paper, we have corrected some explanations in lines 78-80. Theoretically, it may be very possible that the interaction between the two proteins may affect the proteolytic activity of the protease so that ASC may be hydrolyzed. The suggestion of the reviewer is very meaningful, and we would like to conduct explorations in this aspect in the future. Comments 8: ASC oligomerization and ubiquitination are not described in Introduction. Response: The suggestion of the reviewer is very valuable and thanks a lot. We have added a description about ASC oligomerization and ubiquitination in the Introduction in lines 62-65.
Comments 9: Line 81: What is RIMP-1640? Probably, rich culture medium RPMI 1640? Response: We sincerely thank the reviewer for the kind remarks. We apologize for the mistake and we have corrected the words as RPMI 1640 in line 88 and line 133.
Comments 10: Line 122: Origin of all tissue cultures (HEK-293T cells, BEAS-2B cells and THP-1) should be added. Response: We sincerely thank the reviewer for the kind remarks. We have corrected this part in lines 129-131.
Comments 11: What stands for P/S? Antibiotics or something else. It's hardly possible to guess. Response: We sincerely thank the reviewer for the kind remarks. We have added the whole names of P/S in line 133.
Comments 12: Construction of the recombinant plasmids is not complete. Isolation of original SARS-CoV-2 RNA and reverse transcription with subsequent PCR with unknown primers are not described at all. Response: Thanks sincerely for the helpful remarks. We have corrected this part in lines 140-143.
Comments 13: Why PLP-TM (transmembrane?). Is it a truncated gene and recombinant protein fragment or full-length protein? Response: We sincerely thank the reviewer for the kind remarks. PLP-TM (transmembrane) is the amino acid sequence from aa1564 to aa2394 in pp1ab encoded by the DNA sequences of SARS-CoV-2 Wuhan-Hu-1 (GenBank accession number NC_045512.2), which contains PLP domain together with TM domain. According to previous research (Journal of virology 2004, 78 (24), 13600-12), the transmembrane domain may play an important role in the intracellular localization of PLP, so the encoding sequence of PLP-TM was synthesized and used in this study.
Comments 14: Line 140: Structures of primers specific to human pro-IL-1β also remain unknown. Response: We sincerely thank the reviewer for the kind remarks. We have added the specific primers to human pro-IL-1β in lines 147-149. The specific primers are as follows, forward primer GAAT G GATCC GCCACC ATGGCAGAAGTACCTGAG, reverse primer: GCCG G AATTC GGAAGACACAAATTGCATGGTGAAG).
Comments 15: What promoter(s) control transcription of the cloned genes in the transfected eukaryotic cells? Probably, schemes of cloning and subsequent subcloning would be helpful. Response: We sincerely thank the reviewer for the kind remarks. We apologize for the mistakes we made in this part. Now we have corrected this part in lines 149-151. The pro-IL-1β was encoded into pcDN3.1 Flag-tagged vector between BamHI and EcoRI and the pcDN3.1 Flag-tagged vector contains CMV promoter.
Comments 16: Line 144: Presentation of methods as "5E+8 TU/ml" is not common and is not acceptable. Response: We sincerely thank the reviewer for the kind remarks. We have corrected 5E+8 into 5×108 in lines 155-156. The lentivirus used in this study was constructed by Genechem (Shanghai), and “TU/ml” is used as a unit of activity titer of lentivirus. The reason that the active titer unit could not be accurately converted to the physical titer (copies), we have to use its initial unit as “TU/ml”.
Comments 17: Line 164: Lactate dehydrogenase (LDH) is a cytoplasmic enzyme found inside all living cells. After the plasma membrane is damaged, LDH is released into the cultural medium. But cytotoxicity may be without leakage of enzyme(s) through pores of cellular membranes. Therefore, other methods based on mitochondrial reductase that is active in the living cells only (MTT, XTT, WST) are widely used to estimate cytotoxicity. Response: We sincerely thank the reviewer for the kind remarks. As the reviewer mentioned, the MTT, XTT, and WST assays are widely used to estimate cytotoxicity. In our reports, the aim of LDH releasing assay is to reflect cell pyroptosis, as some researchers thought that the release of LDH is an indication of pyroptotic cell death. (1. Shi J, Zhao Y, Wang K, et al. Cleavage of GSDMD by inflammatory caspases determines pyroptotic cell death. Nature. 2015;526(7575): 660-665. doi:10.1038/nature15514. 2. Zheng Z, Bian Y, Zhang Y, Ren G, Li G. Metformin activates AMPK/SIRT1/NF-κB pathway and induces mitochondrial dysfunction to drive caspase3/GSDME-mediated cancer cell pyroptosis. Cell Cycle. 2020;19(10):1089-1104. doi:10.1080/15384101.2020.1743911). So we detected the release of LDH to indicate pyroptotic cell cytotoxicity. In the revised manuscript, we have corrected the expression of “cell cytotoxicity” into “LDH releasing” in method 2.7, results 3.2(lines 284-286), and Figure 3F.
Comments 18: What means "technical replicates"? Response: We sincerely thank the reviewer for the kind remarks. “Technical replicates” used in this paper means the experiments were replicated three times technically.
Comments 19: Results. Figure 1. Abbreviations are not determined. Standard deviations shown as vertical black lines do not correspond to multiple stars that show "the significance of the difference between conditions. *P<0.05, ** P<0.01, *** P<0.001, **** P<0.0001. Response: We sincerely thank the reviewer for the kind remarks. We apologize for the misunderstanding caused by the lack of a clear description at the beginning. All of the boxplots in Figure 1 were plotted using “geom_boxplot” in ggplot2 (ggplot2: Elegant Graphics for Data Analysis. Springer-Verlag New York, 2016) R package. The horizontal line within each box acts as the median, and the bottom and top of each box indicate the 25th and 75th percentile. Wilcoxon rank-sum test was applied to test the significance of the difference between conditions using the “geom_signif” function in ggsignif (Significance Brackets for 'ggplot2') R package.
Comments 20: Figure 2 is multipanel. However, the legend corresponds to the upper luciferase activity assay. Moreover, even for luciferase activity the significance of 4 right columns marked with "+ SARS-CoV PLP" is unclear. What difference among them? PLP from the SARS-CoV or from SARS-CoV-2? Response: We sincerely thank the reviewer for the kind remarks. We have corrected the labels in Figure 2 and added more details in Figure 2’s legends. SARS-CoV PLP was used as a positive control in Figure 2, as in previous reports, SARS-CoV PLP has been found to negatively regulate IFNβ and NF-κB pathway. (1. Devaraj SG, Wang N, Chen Z, et al. Regulation of IRF-3-dependent innate immunity by the papain-like protease domain of the severe acute respiratory syndrome coronavirus. J Biol Chem. 2007;282(44):32208-32221. doi:10.1074/jbc.M704870200. 2. Yang X, Chen X, Bian G, et al. Proteolytic processing, deubiquitinase and interferon antagonist activities of Middle East respiratory syndrome coronavirus papain-like protease. J Gen Virol. 2014;95(Pt 3):614-626. doi:10.1099/vir.0.059014-0). Comments 21: Figure 3 seems overloaded and without a detailed description in the legend is unclear. Response: We sincerely thank the reviewer for the kind remarks. We apologize for the inaccurate description and we have corrected this part and added more details in the legends of figure3.
Comments 22: Figure 5 is mentioned in the text before figure 4. It must be corrected. Response: We sincerely thank the reviewer for the kind remarks. We checked and corrected the order of quotations of Figure 4 and Figure 5 in the text carefully.
Comments 23: Schemes of co-immunoprecipitation and cross-linking could make clear possible evidences of so called "interaction" of protease with protein and oligomerization. Additional evidence might be electrophoresis in native conditions without previous denaturation of samples. Response: We would like to thank reviewer for the very constructive suggestions. These comments inspired us a lot. Co-immunoprecipitation is one of the common methods to indicate the possible interaction of proteins and could only provide possible evidences of protein-protein interaction. According to the valuable suggestion of the reviewer, we have changed the expression of “interaction” into “coimmunoprecipitate” and hope that it might be more precise. To add more evidence, we conducted immunofluorescence assay to uncover the co-localization of PLP-TM and ASC in cells. As the results below, SARS-CoV-2 PLP partially co-localizes with ASC in cells.
Comments 24: Figure 5, part A.Scale bars are without numbers. Results of confocal fluorescent microscopy should be shown with the scale bars and description of magnification in the legend. Response: Thanks for the helpful remarks. We have corrected this part according to your requirements.
Comments 25: What difference between specks and fluorescent micro- or nanoparticles? Response: Thanks for the helpful remarks. ASC specks were first described in apoptotic cells in 1999 (Masumoto J, Taniguchi S, Ayukawa K, et al. ASC, a novel 22-kDa protein, aggregates during apoptosis of human promyelocytic leukemia HL-60 cells. J Biol Chem. 1999;274(48):33835-33838. doi:10.1074/jbc.274.48.33835). Upon triggering of inflammasome sensors, ASC assembles into large helical fibrils that interact with each other serving as a supramolecular signaling platform termed the ASC speck (Hoss F, Rodriguez-Alcazar JF, Latz E. Assembly and regulation of ASC specks. Cell Mol Life Sci. 2017;74(7):1211-1229. doi:10.1007/s00018-016-2396-6). While nanoparticles, according to the American Society of Testing Materials (ASTM) standard definition, are particles with lengths that range from 1 to 100 nanometers in two or three Dimensions (Lewinski N, Colvin V, Drezek R. Cytotoxicity of nanoparticles. Small. 2008;4(1):26-49. doi:10.1002/smll.200700595). The differences between ASC specks and fluorescent micro- or nanoparticles are that ASC specks are the spontaneous formations after inflammatory activation. |
||||
4. Response to Comments on the Quality of English Language |
||||
Point 1: Extensive editing of the English language required |
||||
Response 1: Thanks for your suggestions on the quality of the English language, we have polished the language according to the weblink the editor provided. All the modifications were highlighted in the revised manuscript. |